# Investigating the Effects of Intraprofessional Learning in Nursing Education: Protocol for a Longitudinal Study

Debra Kiegaldie [1,2,3], Ishanka Weerasekara [4,5,6] and Louise Shaw [1,7,*]

1    Faculty of Health Science, Youth and Community Studies, Holmesglen Institute, 488 South Road, Moorabbin, VIC 3189, Australia
2    Eastern Health Clinical School, Monash University, Clayton, VIC 3128, Australia
3    Healthscope, Holmesglen Private Hospital, 488 South Road, Moorabbin, VIC 3189, Australia
4    Faculty of Health and Social Sciences, Western Norway University of Applied Sciences, 5063 Bergen, Norway
5    School of Health Sciences, The University of Newcastle, University Drive, Callaghan, NSW 2308, Australia
6    School of Allied Health Science and Practice, Faculty of Health and Medical Sciences, The University of Adelaide, Adelaide, SA 5005, Australia
7    Academic and Research Collaborative in Health (ARCH), School Allied Health, Human Services and Sport, La Trobe University, Bundoora, VIC 3086, Australia
*    Correspondence: louise.shaw@latrobe.edu.au

**Abstract:** Interprofessional learning (IPL), where nursing students learn how to work with multiple health professionals in their future practice to deliver the highest quality of care, has become an essential feature of undergraduate nursing programs. Intraprofessional learning (IaPL) is where individuals of two or more disciplines within the same profession collaborate; however, there is a dearth of literature investigating its effects in nursing education. The aim of this study is to investigate the impact of IaPL on the development of nursing students' knowledge, skills, and attitudes for collaborative practice. The study will utilize a mixed methods approach with surveys conducted at six time points across two years of two nursing programs and focus groups at the end of the program. Participants will be recruited from the Diploma and Bachelor of Nursing programs at an Australian Training and Further Education institute. Four specific IaPL educational experiences incorporating simulation will be developed on aged care, mental health, complex care and acute care. The study will provide nursing students with multiple opportunities to develop the necessary capabilities for collaborative practice. It will longitudinally evaluate nursing students' attitudes towards IaPL and examine whether IaPL motivates Diploma of Nursing students to pathway into a Bachelor of Nursing degree. The study will also investigate awareness amongst nursing students of the scope of practice, roles and responsibilities of the nursing team.

**Keywords:** education; nursing; intraprofessional learning; simulation; simulation-based education

## 1. Introduction

With a long history in healthcare education, interprofessional learning (IPL) has become an essential feature of undergraduate nursing programs. Interprofessional learning in this context, is defined as when two or more healthcare students learn with, from and about each other [1]. In nursing education, the intent of IPL is for nurses to learn how to work with multiple health workers from different professional backgrounds to deliver the highest quality of care to patients, carers, families and communities [2]. Intraprofessional learning (IaPL) is the learning that occurs when individuals of two or more disciplines within the same profession collaborate [3,4]. Interprofessional learning and its positive effects are widely described in the literature [3]. Whilst IaPL is less well documented in health professional education, it has been reported in some studies, for example, dentistry students and dental hygienists learning collaboratively to deliver team-based care [5–10], occupational therapy and occupational therapist assistant students learning together [11],

physical therapists and physical therapy assistants collaborating within the classroom [12] and intraprofessional workplace learning in postgraduate medical education [3].

Within nursing, a healthy clinical environment is supported by collaboration between colleagues, and has been reported to improve job and patient satisfaction and ensure high-quality nursing care [13]. IaPL plays a pivotal role in undergraduate education by equipping student nurses to learn how to work together as a cohesive nursing team, using effective intraprofessional communication and collaborative skills [14,15]. However, collaborative learning between nursing students is researched in few studies and therefore there is a critical need to conduct outcomes-based research in this area. The few existing studies that report the effect of IaPL in pre-registration nursing education courses are mostly from the USA or Canada (for example, [16–21]) but none are from Australia. To enhance collaborative nursing care, nursing education that emphasizes role clarity, promotes communication and develops scope of practice via various shared learning experiences, is required [16,20,21]. The maintenance of siloed education of degree and diploma nursing courses was found to perpetuate professional boundaries and hierarchical structures [16,20]. Barriers created by siloed nursing education were also found to contribute to a lack of awareness of each other's roles amongst registered and practical nurses post-graduation, which inhibited collaborative nursing practice [22].

A range of innovative teaching methods have been used across the health professions in IPL including nursing, dentistry, physiotherapy, occupational therapy and medicine [5,6,12,14,15,23,24]. These include simulation-based education (SBE), student-led training wards, problem-based learning and group projects [25,26]. SBE offers an ideal learning environment for students to learn the skills required to effectively engage with patients and other health care providers and is a commonly used intervention to introduce and assess interprofessional collaboration [6,15,24,27–29]. SBE has been used to teach IaPL to nursing students in a few studies [14,17,19,27,28]. For example, participation in a school-based clinical telehealth simulation increased nursing students' readiness for intraprofessional collaboration [19]. However, it was suggested that future studies should sample learners earlier in their training and in multiple implementation sites [19]. Boothby et al. (2019) found that students participating in intraprofessional simulations initially reported some feelings of discomfort and intimidation [17]. However, with greater preparation of students and pre-briefing, students reported the simulations helped to developed teamwork, collaboration and communication [17].

There is no published validated instrument to assess the intraprofessional experience to date. The adaptation of instruments that measure attitudes towards interprofessional learning within the IaPL context is virtually non-existent. Further, the factors that support and facilitate the implementation of IaPL in tertiary settings, (i.e., universities or vocational training institutions) have not been comprehensively assessed [30]. The effects of intraprofessional learning in nursing education and the views and experiences of students, nursing faculty and industry clinical educators are yet to be examined in Australia.

This project aims to investigate the attitudes of nursing students towards intraprofessional learning using simulation at an Australian Training and Further Education (TAFE) institute, and whether intraprofessional learning motivates Diploma of Nursing students to pathway into a Bachelor of Nursing degree. The views of nursing faculty in facilitating intraprofessional learning and industry clinical educators on the application of IaPL in the clinical setting will also be explored. In addition, the study will determine the validity and reliability of an adapted interprofessional learning instrument to the intraprofessional learning context.

## 2. Materials and Methods

### 2.1. Design

This study will adopt a longitudinal cohort study design with six data collection points. It will commence January 2023 with students from the Diploma and Bachelor of Nursing courses at a TAFE institute and will follow two years of their studies. It will conclude with a data collection point 6 months post-graduation.

### 2.2. Ethics

All methods will be conducted in accordance with the principles of the Declaration of Helsinki and Good Clinical Practice guidelines. Informed consent to participate in the study will be obtained from eligible participants prior to study enrollment. The study protocol was approved by Holmesglen's Human Research Ethics Review Panel (04/2022). The researcher will obtain written informed consent from eligible participants.

### 2.3. Participants and Recruitment Methods

**Students:** This project will recruit eligible student participants from the Diploma and Bachelor of Nursing programs at the Australian TAFE institute who have enrolled for the year 2023–2024. These participants will be followed until 2025.

Bachelor of Nursing (BN): This is a three-year undergraduate degree that runs over four stages. The study will involve students from year two who will be followed until six months post-graduation. Approximate number of eligible students will be n = 300.

Diploma of Nursing (DN): This is a two-year course that runs over four stages. The study will involve first stage DN students who will be followed until six months post-graduation. Approximate number of eligible students will be n = 400.

**Nursing faculty:** Nursing faculty (approximately n = 30) for the DN and BN courses will be involved in facilitating the simulated intraprofessional interventions. They will be invited to take part in the study to explore their views and experiences of delivering the IaPL interventions across the two years.

**Industry clinical educators:** Industry clinical educators (approximately n = 10) from partnering health services who supervise DN and BN students and graduates will be invited to take part in the study to explore their views and experiences of students and graduates who have completed IaPL interventions.

### 2.4. Recruitment

**Students and Nursing faculty:** Following advertisement of the program, students and nursing faculty will be invited to participate in the research via an email sent from the faculty office. The participant information will be attached to the email. Students and faculty will be advised they can choose to not participate in the study without any negative consequences. The email will include a link which will take them to an online survey using the Qualtrics™ (Seattle, WA, USA) platform. Students or faculty can then mark a checkbox in the survey to indicate their consent to participate in the survey(s) and to participate in an audio-recorded focus group interview. The focus group interview will take place either in person or online at a mutually convenient date and time. Reminders will be sent a day before the scheduled time.

**Industry clinical educators:** Industry clinical educators will be recruited to join a focus group interview via an email from the faculty office with the participant information attached. Faculty will be advised they can choose to not participate in the study without any negative consequences. The email will include a link where project team members can mark a checkbox to indicate their consent. The focus group interview will take place either in person or online at a mutually convenient date and time. Reminders will be sent a day before the scheduled time.

*2.5. Setting*

The setting for the study will be the classrooms of the TAFE institute.

*2.6. Intervention*

The program involves the delivery of four Intraprofessional Learning Modules to develop students' knowledge, skills and attitudes towards intraprofessional learning and collaborative practice. The learning modules will incorporate ice breakers, team-based small group learning, simulated clinical scenarios and a range of interactive intraprofessional learning activities. The interventions will take place over 2 years, across four semesters.

Module 1: Aged Care—The Cognitively Impaired Patient
Module 2: Mental Health—The Transgender Client
Module 3: Complex Care—Palliative Care
Module 4: Acute Care—The Deteriorating Patient

Healthcare simulation standards of best practice will be utilized [31]. Each module will include a simulation scenario involving a briefing, immersive simulation experience followed by debriefing and feedback according to the PEARLS framework [32]. All teaching activities will be facilitated by pairs of experienced DN and BN faculty who have received training.

The modules and associated simulated clinical scenarios have been developed by the research and teaching teams of the TAFE institute.

*2.7. Data Collection*

**Students:** A mixed methods approach will be utilized consisting of pre- and post-test surveys and focus group interviews. Students will be asked to complete the surveys online using Qualtrics™ (Seattle, WA, USA) at six different time points:

1. Demographic information collected at Time Point 1 [T0].
2. The modified KidSIM$^{TM}$ Attitudes Towards Teams in Training Undergoing Designed Educational Simulation Questionnaire (M-ATTITUDES), to suit the IaPL context. All 6 time points [T0–T5].
3. Self-efficacy questions to elicit views on knowledge, motivation and confidence for intraprofessional collaborative practice at 5 time points [T0–T5, confidence items only for T0].
4. Learning experience questions to elicit views about the intraprofessional teaching approaches at 4 time points [T1–T4].
5. For Diploma of Nursing students, there will be a question about whether the IaPL increased their motivation to enroll in a Bachelor of Nursing degree in the future at 5 time points [T1–T5].
6. One open-ended question asking for general comments on the IaPL at 5 time points [T1–T5].

The baseline survey will be distributed to students online before their first IaPL experience. An invitation to complete the survey will be sent to their student email address. The next four post-intervention surveys will be distributed at the conclusion of each IaPL experience. The final survey will be distributed 6 months post-graduation using an email address provided by participating students on graduation. The data will be collected between February 2023 and June 2025 (Table 1 and Figure 1).

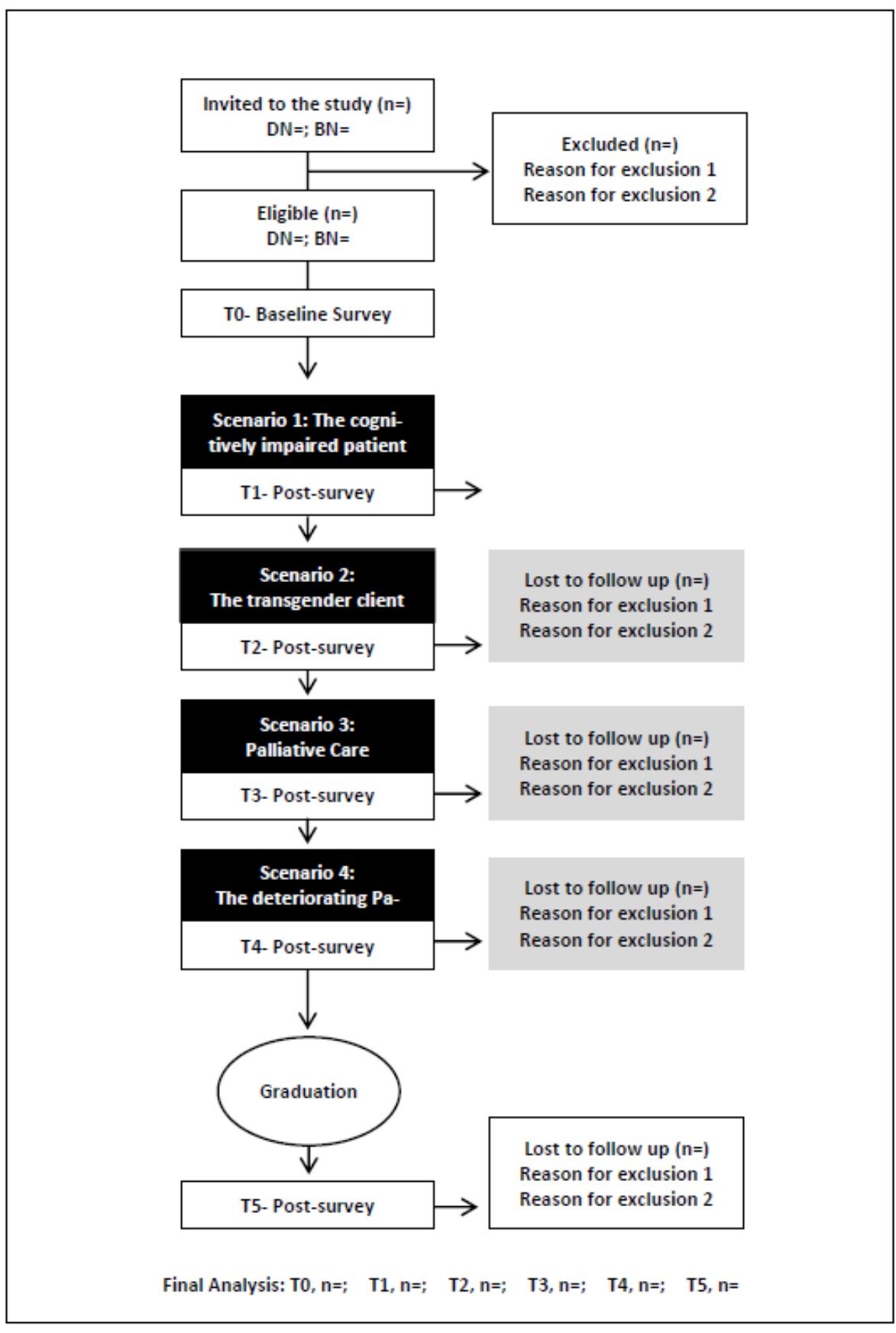

**Figure 1.** Study flowchart for student surveys.

**Table 1.** Student data collection protocol.

| Time | Timing | IaPL Intervention | Course | Year | Sem | Type of Data |
|------|--------|-------------------|--------|------|-----|--------------|
| T0 | 02/2023 | Pre-test | BN | 2 | 1 | **Survey 1**<br>• Demographics<br>• M-ATTITUDES<br>• Self-efficacy questions (confidence only) |
|   |   |   | DN | 1 | 1 |   |
| T1 | 06/2023 | Aged Care—The Cognitively Impaired Patient | BN | 2 | 1 | **Survey 2**<br>• M-ATTITUDES<br>• Self-efficacy questions<br>• Learning experiences<br>• FGI 1 |
|   |   |   | DN | 1 | 1 |   |
| T2 | 11/2023 | Mental Health—The Transgender Client | BN | 2 | 2 | **Survey 3**<br>• M-ATTITUDES<br>• Self-efficacy questions<br>• Learning experiences<br>• FGI 2 |
|   |   |   | DN | 1 | 2 |   |
| T3 | 06/2024 | Complex Care—Palliative Care | BN | 3 | 1 | **Survey 4**<br>• M-ATTITUDES<br>• Self-efficacy questions<br>• Learning experiences<br>• FGI 3 |
|   |   |   | DN | 2 | 1 |   |
| T4 | 11/2024 | Acute Care—The Deteriorating Patient | BN | 3 | 2 | **Survey 5**<br>• M-ATTITUDES<br>• Self-efficacy questions<br>• Learning experiences<br>• FGI 4 |
|   |   |   | DN | 2 | 2 |   |
| T5 | 06/2025 | Final Post-test 6 months post-graduation | BN | N/A | N/A | **Survey 6**<br>• M-ATTITUDES<br>• Self-efficacy questions |
|   |   |   | DN | N/A | N/A |   |

Focus Groups

Each focus group will take approximately 1 h and will be audio-recorded and transcribed.

**Students:** After delivery of each intraprofessional scenario, consenting students will be invited to take part in a semi-structured focus group interview where they will be asked to share their views on the IaPL experience, its impact on achieving the learning outcomes and changes to their attitudes towards IaPL.

**Nursing faculty:** On completion of the program, a selection of consenting nursing faculty who delivered the Intraprofessional Learning Program will be invited to a focus group interview. Questions will be focused on their views and experiences of facilitating intraprofessional teaching activities.

**Industry Clinical Educators:** On completion of the program, a selection of consenting industry clinical educators who supervise nursing students or graduates of Holmesglen who have received the intervention will be invited to a focus group interview. Questions will be focused on their views of intraprofessional practice and intraprofessional attitudes and behaviors of nursing students or graduates.

The timeline for all study activities can be viewed in Figure 2.

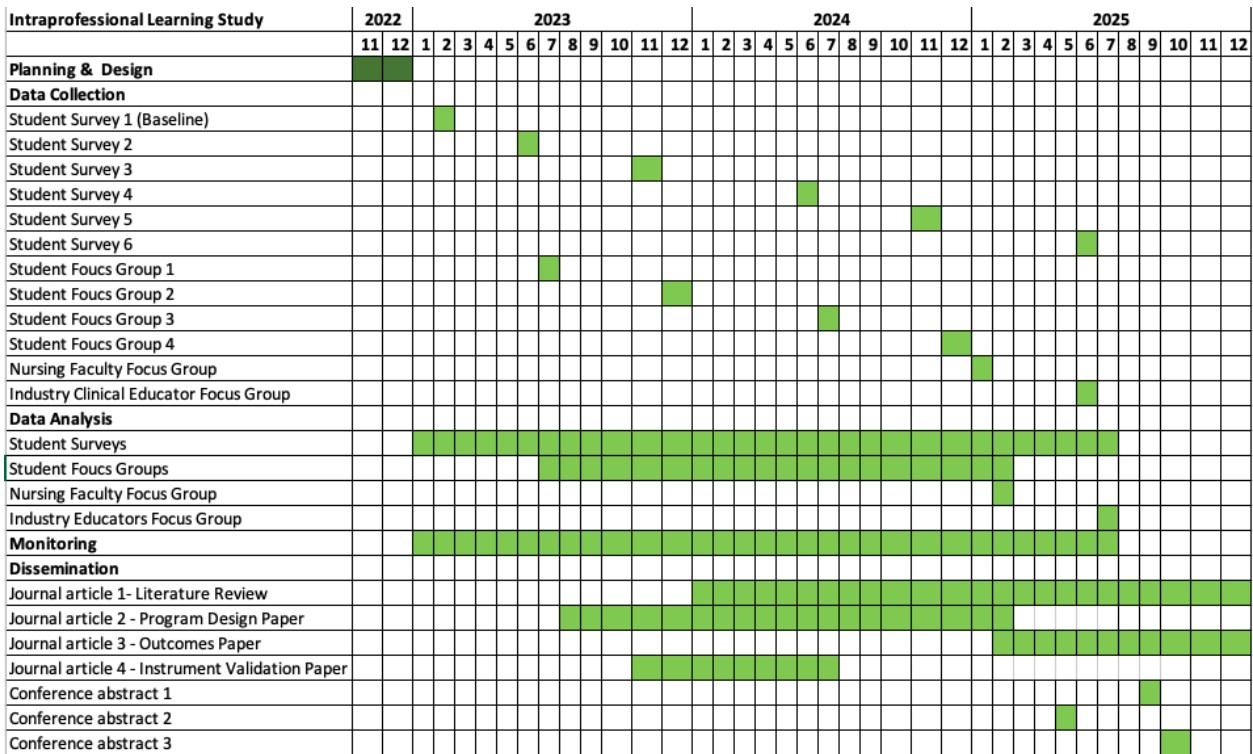

**Figure 2.** Gantt chart for research timeline.

*2.8. Instruments*

Two types of instruments will be used in this study:

Student Surveys

Participant Characteristics: A small number of questions will identify student characteristics such as campus, course, semester, gender, enrollment status (domestic/international) and age.

The M-ATTITUDES Survey: A meaningful investigation of the intraprofessional learning environment needs a valid and reliable instrument, yet no instrument has been found to reliably assess the intraprofessional learning experience. Existing attitude scales on interprofessional education (IPE) focus on students' attitudes towards concepts of teamwork and opportunities for IPE but fail to examine student perceptions of the learning modality that also plays an important role in the teaching and learning process [33]. The original design of the ATTITUDES survey was to measure student perceptions of, and attitudes toward IPE, teamwork and simulation as a learning modality [33]. It has been tested in several settings and has been found to be a reliable and valid measure.

ATTITUDES contains 30 items on a 5-point Likert scale from Strongly Disagree = 1 to Strongly Agree = 5. The original factor analysis supported a 5-factor solution accounting for 61.6% of the variance: communication (8 items), relevance of IPE (7 items), relevance of simulation (5 items), roles and responsibilities (6 items) and situation awareness (4 items). Excellent internal reliability was reported as a = 0.95 [33].

ATTITUDES has been used in several studies since 2012 in the context of interprofessional education [34–39]. Sanko et al. used a modified version of ATTITUDES with 28 items, before and after a week-long interprofessional simulation-based patient safety course [37]. Shanahan and Lewis used ATTITUDES before and after a clinical simulation-based interprofessional education exercise [39]; however, this study did not explicitly explain the methods. ATTITUDES was used by El- Gamal et al. in 2017 aiming to investigate the perceptions and attitudes of critical care students towards the simulation [35]. They modified the tool by replacing the interprofessional education team with teamwork

and team leader roles and responsibilities. In the same year Seale et al. used ATTITUDES before and after their high-fidelity simulation [38]. Recently, ATTITUDES was used as a pre-post survey allowing students to share detailed thoughts and feelings about the mass casualty simulation experience by James et al. [36]. AlBalawi et al. (2022) [34], used ATTITUDES during their simulation-based education experience to measure students' attitudes and beliefs about their simulation training. The tool will be modified to suit the IaPL context (M-ATTITUDES).

Self-efficacy questions: From Survey 2, three self-efficacy questions (designed by the researchers) will ask students to rate on a 5-point rating scale (1 = much worse now, 5 = much better now) their perceptions of the effects of the intraprofessional learning experiences on their knowledge of intraprofessional collaborative nursing practice, their knowledge of roles and scope of practice of enrolled and registered nurses, and their motivation to collaborate intraprofessionally. Students will also be asked to rate their confidence in demonstrating intraprofessional nursing practices linked to the learning objectives of the IaPL modules (4 items).

Views about intraprofessional teaching: This consists of eight questions on a 5-point Likert scale (1 = strongly disagree, 5 = strongly agree) asking students their views about the intraprofessional learning experiences.

Additional questions: Two additional questions for DN students will ask whether the intraprofessional learning experience has increased their motivation to enroll in a Bachelor of Nursing degree in the future and ask them to explain how this experience helped their decision. A final question will ask for any additional comments. Survey 6 will include two questions to identify the type of health care settings and area of nursing that the graduate is currently working in.

### 2.9. Data Analysis

**Quantitative data:** Demographics will be analyzed using frequency counts and percentages. These results will be compared to the group as a whole and reported as both numbers and percentages. For the M-ATTITUDES (across 6 surveys), descriptive statistics will be calculated using the Statistical Package for Social Sciences (SPSS) Version 27 (IBM SPSS Statistics for Windows, Armonk, NY, USA) for each item, overall scale scores and sub-scale scores. Paired sample $t$-tests will be used to assess for differences between the participants pre- and post-test across all 6 time points. An analysis of variance (ANOVA) will be used to look at statistical differences in scoring overall then according to course (DN and BN), age and enrollment status (domestic and international). Reliability coefficients (Cronbach $\alpha$) will be used to assess for internal consistency of the sub-scales and total survey. The same process will be applied to the Self Efficacy questions (across 5 surveys).

For the survey items on the teaching process, summary results of frequencies (percentages and numbers), mean, standard deviation, standard error of the mean and 2X SEM will be calculated for all items. All items will be analyzed and compared according to course (DN and BN), age and enrollment status (domestic and international).

In this study we will test psychometric properties of the M-ATTITUDES survey in our cohort of DN and BN nursing students in 2024 using SPPS. A principal components analysis with varimax rotation will be completed to assess the validity of the constructs being measured and the variance attributed to each of the underlying factors. The resulting eigenvalues that are greater than 1 will be used to identify the number of factors. Items loading more than 0.32 will be identified and assigned according to where they loaded the highest on 1 of the 5 factors. An analysis of the international reliability (Cronbach $\alpha$) will be completed for each of the resulting sub-scales factors and the total survey. Paired sample $t$-tests will be used to look at mean differences from Survey 1 to Survey 6 for each factor or subscale.

**Qualitative data:** Survey data on open-ended responses will be transcribed and qualitative data will be analyzed and reported thematically using Excel. Two researchers will code the data and any tenable strategies to achieve trustworthiness of data will be implemented.

### 2.10. Risk Management and Safety

It is anticipated that there will be no physical, psychological, social, legal or financial harm to the participants involved in this study. Participants may withdraw from the study at any time. Students will be informed that participation is voluntary and their educational standing will not be impacted by whether they choose to participate or not. No identifying information will be included in any survey and identifying information will be redacted from focus group data.

### 2.11. Data Security and Handling

Survey data will be collected online and saved in a password protected Qualtrics™ account. Other information (consent form) will be collected and stored on the institute's secure drive, which only the researchers have access to via their institution login and password. Focus groups will be recorded and transcribed by an administrator. These recordings will only be available to the researchers via use of a password. Any identifying information will be removed from the transcripts being analyzed.

### 2.12. Confidentiality and Security

All information collected will be anonymous and no individual will be identifiable in any of the reporting of outcomes. Information will not be discussed or shared with anyone other than the researchers.

Students will be required to enter their student ID on each survey for the purposes of avoiding duplicate responses and for pairing of all surveys for statistical analysis. Once paired, the ID is re-identified or 'recoded' and all student IDs are removed. All identifying features of the student ID will be re-identified and only aggregated data will be reported.

Findings of the research will be presented in a de-identified manner in published peer-reviewed journals and presented at conferences. It will not be possible to identify students from any information presented in this manner.

During the study, all files will be kept secure for the duration of the project. Following completion of the study, project documentation will be kept in a secure, password protected location on a Holmesglen drive. Data will be stored for 5 years, and then will be discarded securely.

### 3. Conclusions

This study aims to evaluate the design, implementation and outcomes of an IaPL program for nursing students. It will measure the impact of intraprofessional learning on intraprofessional collaborative practice. It will determine whether IaPL enables students to work collaboratively post-graduation and if it facilitates awareness of each other's roles. Given the infancy of research into IaPL, the study represents a unique example of how to design effective intraprofessional learning educational experiences for nursing students. In addition, it will expand the evidence base for IaPL in nursing education and provide a research template that could be replicated in other countries and contexts.

The findings of this study will be disseminated in peer-reviewed journals and via national and international conferences from within and outside the partnering institutions' countries. We propose that our approach is an innovative way to prepare a collaborative nursing workforce.

**Author Contributions:** L.S., D.K. and I.W. drafted the manuscript. I.W., D.K. and L.S. are actively involved in the study. All authors have read and agreed to the published version of the manuscript.

**Funding:** This research received no external funding.

**Institutional Review Board Statement:** The study will be conducted in accordance with the Declaration of Helsinki and approved by Holmesglen's Human Research Ethics Review Panel (HRERP), with the project number: 04/2022.

**Informed Consent Statement:** Informed consent will be obtained from all subjects involved in the study.

**Data Availability Statement:** The data that support the findings of this study will be available from the corresponding author, L.S., upon reasonable request.

**Acknowledgments:** We would like to thank Holmesglen Institute's students and nursing faculty for their participation. We would also like to thank industry clinical educators for their participation.

**Conflicts of Interest:** The authors declare no conflict of interest.

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
