# Peer review of "Investigating the Effects of Intraprofessional Learning in Nursing Education: Protocol for a Longitudinal Study"

_nursrep, doi:10.3390/nursrep13020065_

Round 1

Reviewer 1 Report

  The designed study is very valuable for increasing the quality of nursing education and preparation for work in interdisciplinary teams. The project is very interesting and innovative, but the results and conclusions drawn will be more interesting than the research idea. I believe that for the sake of correctness, it would be worth presenting a review of the literature on research that has been carried out on this topic (studies from the USA and Canada) with an indication of the strengths and weaknesses of these studies.

Author Response

Literature on intraprofessional learning in nursing education in the USA and Canada has now been identified and discussed in the introduction. These studies will be discussed further in subsequent publications to compare the findings of our study.

Reviewer 2 Report

“Investigating the effects of intraprofessional learning in nursing education: protocol for a longitudinal study” is very interesting work.

 It addresses a very important topic for clinical field and nursing educators and students.

 The manuscript is well written and well presented.

 Upon my review, I would like to share the following points with the authors:

 1. I recommend revising and editing the layout of Table 1 (data collection) to make it easier to follow and read.

 2. Please modify the scenario in Figure 1 to a module and modify 'Delirium' and 'Mental health' to 'the cognitively impaired patient' and 'the transgender client'. It is suggested because it is different from the module contents in page3.

2. What is M-ATTITUDE in 2.8. Instruments ? What’s meaning the large M ?

 3. Page9, Please mark the qualitative data letters as dark as quantitative data.

Author Response

  • I recommend revising and editing the layout of Table 1 (data collection) to make it easier to follow and read.
    • Response: This table has now been formatted to make it easier to read.
  • Please modify the scenario in Figure 1 to a module and modify 'Delirium' and 'Mental health' to 'the cognitively impaired patient' and 'the transgender client'. It is suggested because it is different from the module contents in page 3.
    • Response: The ‘modules’ are aged care, mental health, complex care and acute care. The scenarios are the cognitively impaired patient etc. The figure has been modified:

      Module 1: Aged Care – The Cognitively Impaired Patient

      Module 2: Mental Health– The Transgender Client

      Module 3: Complex Care - Palliative Care

      Module 4: Acute Care – The Deteriorating Patient
  • What is M-ATTITUDE in 2.8?
    • M = Modified. The sentence now reads, ‘The modified KidSIMTM Attitudes Towards Teams in Training Undergoing Designed Educational Simulation Questionnaire (M-ATTITUDES)…’
  • Page 9, Please mark the qualitative data letters as dark as quantitative data
    • Quantitative and qualitative data letters are now the same colour.

Reviewer 3 Report

It's a very interesting protocol. The proposed methodology is scientifically accepted. The utilization of a non-probability sample of a non-probability sample can be a mistake because it is not representative of the population. As the population is known, it should be considered to use a probability sample. 

Author Response

It's a very interesting protocol. The proposed methodology is scientifically accepted. The utilization of a non-probability sample can be a mistake because it is not representative of the population. As the population is known, it should be considered to use a probability sample. 

Response: In our study we do not plan to randomize the subjects therefore, we can confirm that the sampling method is not a probability sampling method. We agree that having a non-probability sample is not representative of the population. However, we cannot follow this sampling method due to the nature of the recruitment method. Therefore, we will acknowledge this in the limitation section in our future papers produced based on this protocol.  

Round 2

Reviewer 1 Report

Thank you for correcting the manuscript. It is suitable for publication.